# Influence of the Widespread Use of Corten Plate on the Acoustics of the European Solidarity Centre Building in Gdańsk

**Wojciech Targowski** [1] **and Andrzej Kulowski** [2,*]

1   Department of Residential Architecture, Faculty of Architecture, Gdańsk University of Technology,
    ul. G. Narutowicza 11/12, 80-233 Gdańsk, Poland; wojtargo@pg.edu.pl
2   Department of Technical Fundamentals of Architecture Design, Faculty of Architecture, Gdańsk University
    of Technology, ul. G. Narutowicza 11/12, 80-233 Gdańsk, Poland
*   Correspondence: kulowski@pg.edu.pl; Tel.: +48-501-052-874

**Abstract:** This paper describes the relationship between a strong architectural vision that is difficult to balance, and user expectations in terms of acoustics. The focus is on the use of corten steel as the dominant finishing material on façades and interiors to achieve an expressive, symbolic message through program-based design. The architectural premises justifying the adopted solutions are presented, especially the universality and homogeneity of the material. Against this background, the influence of corten steel on the acoustics of the two largest rooms of the European Solidarity Center, which are the winter garden and the multi-purpose hall, was discussed. Remedial steps have been taken to reduce the greatest acoustic inconveniences resulting from the widespread use of metal sheet as a finishing material in rooms, i.e., excessive reverberation and a low degree of sound dispersion. A positive result for the acoustic conditions achieved in the winter garden was the presentation of a large body of classical music in the building.

**Keywords:** symbolic building; corten plates; tilted walls; winter garden; room acoustics; reverberation time; flutter echo

## 1. Introduction

Developed as the result of a high-profile international competition, the European Solidarity Centre (ECS) in Gdańsk was opened in 2014 [1,2]. The purpose of the building is to document and promote the idea of Solidarity, a concept born in the Gdańsk Shipyard, the guiding idea behind the great socio-political changes in Central and Eastern Europe at the end of the 1980s. For understandable reasons, the symbolic aspect of the architecture was decisive for the design solutions and consequently for the verdict of the international jury [3]. The shapes and materials used are closer to sculptural forms than to utilitarian public architecture (Figure 1). A strong formal concept, however, sometimes entails disadvantages in the functional sphere, such as achieving the desired acoustic characteristics.

Widely used as an interior finishing material, the corten plate plays a special role in developing the symbolic code of architecture [4]. Devoid of any paint hiding the essence of the material, the raw surfaces perfectly reflect the spirit of the spontaneous protests by Gdańsk shipyard workers. The corroded surface of the façades, however, is only a complement to the primary formal idea behind the project, referring to the once existing steel stockpile required for shipbuilding—hence the consistent parallel alignment, and the inclination from the upright position of the rust-colored walls, so that they directly relate to the metal sheets leaning against the storage racks (Figure 2).

The aim of this paper is to show the possibility of reconciling the specific, difficult-to-balance, formal program of architecture with the requirements of the acoustic usability of the building (Figure 3).

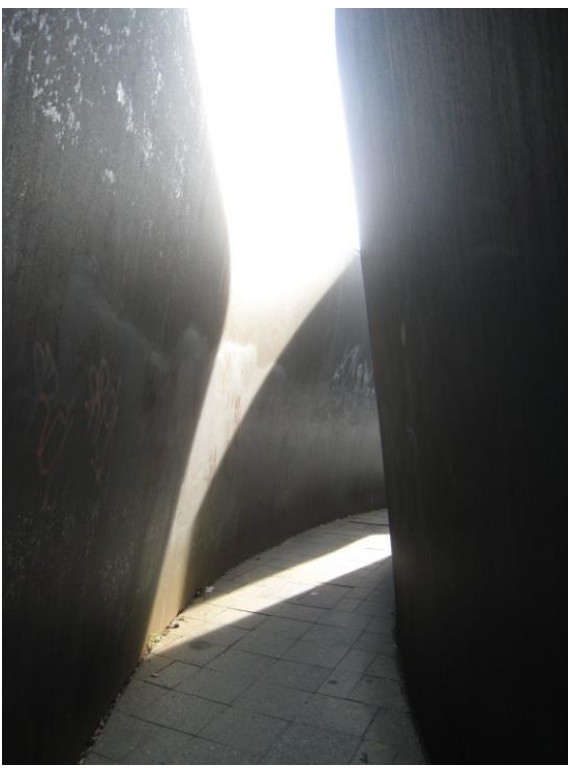

**Figure 1.** Richard Serra, Memorial to the victims of the Nazi T4 extermination program, Berlin. Material: corten plate (photo by Wojciech Targowski).

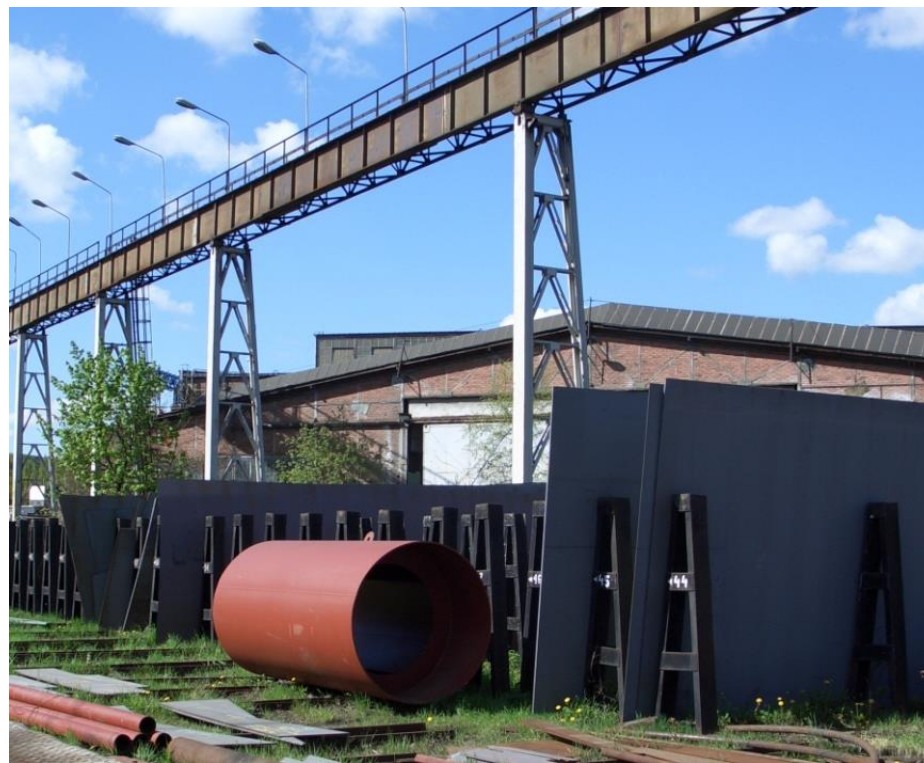

**Figure 2.** Gdansk Shipyard, metal sheets warehouse for shipbuilding on the site of the future European Solidarity Centre (ECS) building (photo by Wojciech Targowski).

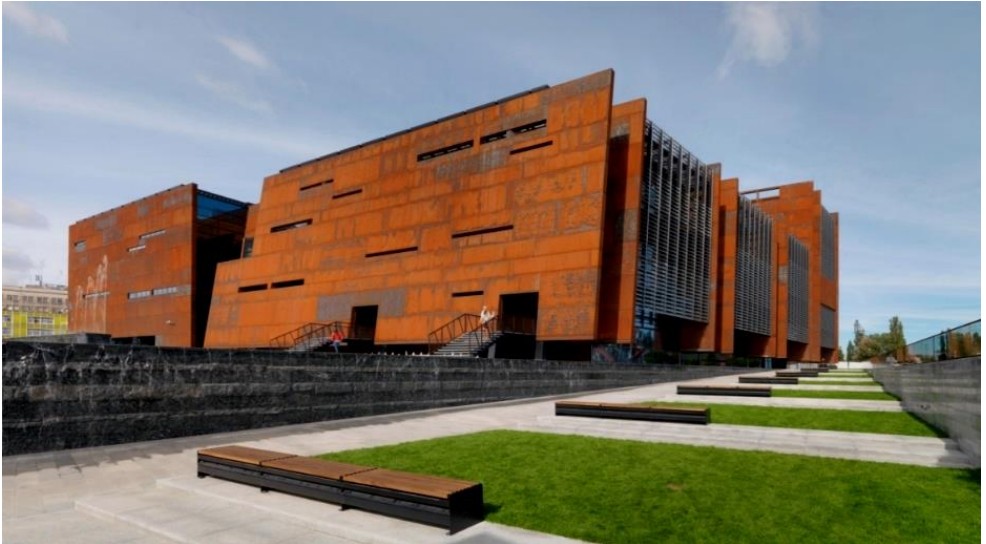

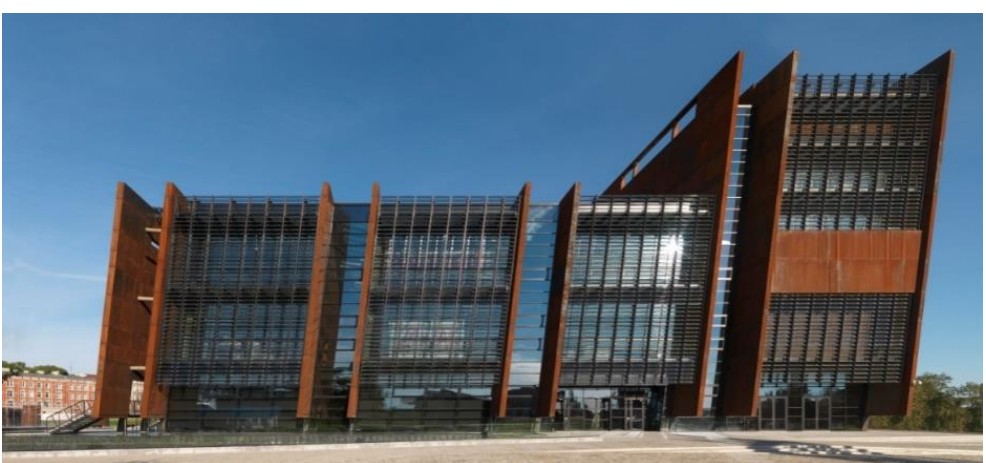

**Figure 3.** The south and west façade of the ECS, with structural walls which deviate from the vertical by 6.5°. The façades of the ECS are covered with 5 mm thick corten plates (photo by Wojciech Kryński).

## 2. Examples of Widespread Use of Corten Cladding in Public Utility Buildings

The use of corten steel in building architectural narratives dates back to the 1960s. One of the precursors of its application was the eminent American–Finnish architect Eero Saarinen. In 1964, after his death, the John Deere World Headquarters designed by him in Moline, Illinois, USA, was put into operation. In the symbolic code of the architecture of this building, an attempt was made to establish a dialogue of seemingly opposing values—industrial civilization and nature. According to the production profile, a large concern producing heavy agricultural machinery has gained a complex of offices and exhibition facilities made of steel blended into the organic landscape [5] (Figure 4).

The Mies van der Rohe award-nominated Raif Dinckok Cultural Center was opened in 2011 in Yalova, Turkey. The use of corten steel in the architecture of the building is an obvious reference to the industrial character of the city. It is also a tribute to the founder—a large industrial company, the Akkok Group. The impressive, openwork, steel structure, illuminated by flickering light, contradicts the perception of steel as a soulless, raw matter and, contrary to stereotypes, encourages the affirmation of the industrial landscape (Figure 5) [7].

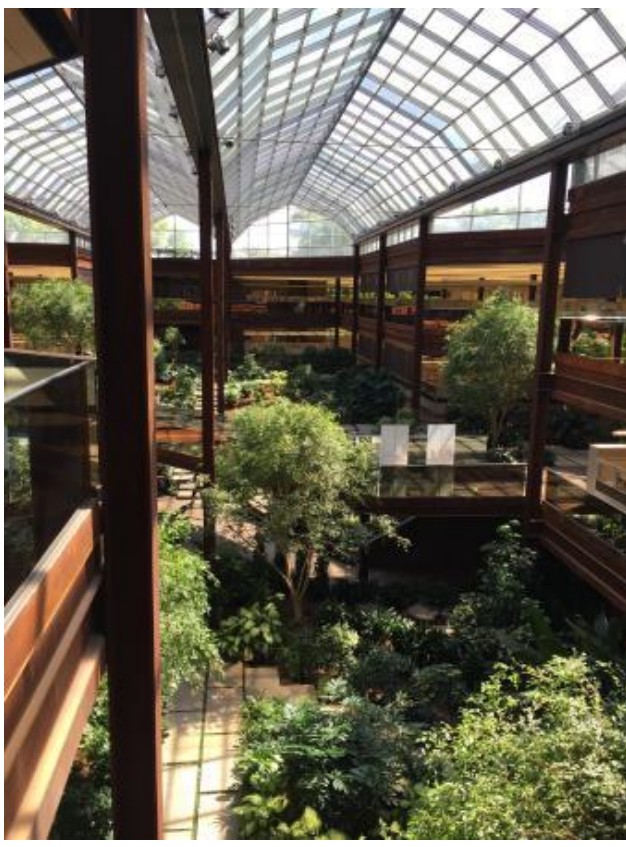

**Figure 4.** John Deere World Headquarters Moline, Illinois, USA. Object completed in 1964, architects—Eero Saarinen and Kevin Roche [6]. Photo—public domain.

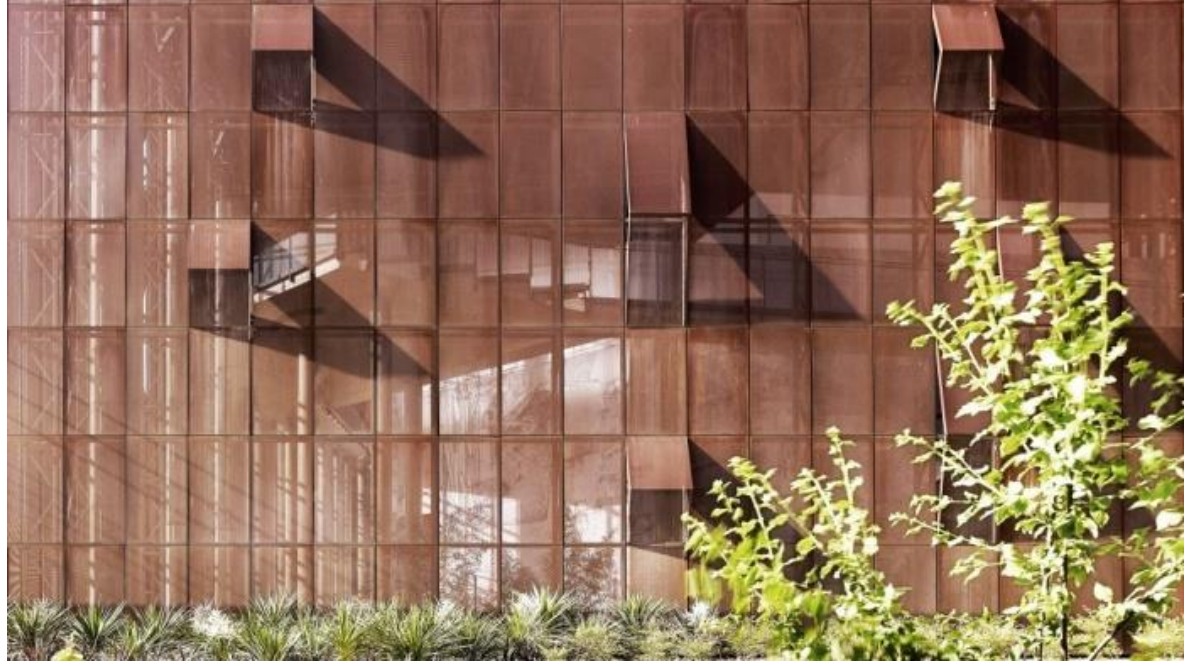

**Figure 5.** *Cont.*

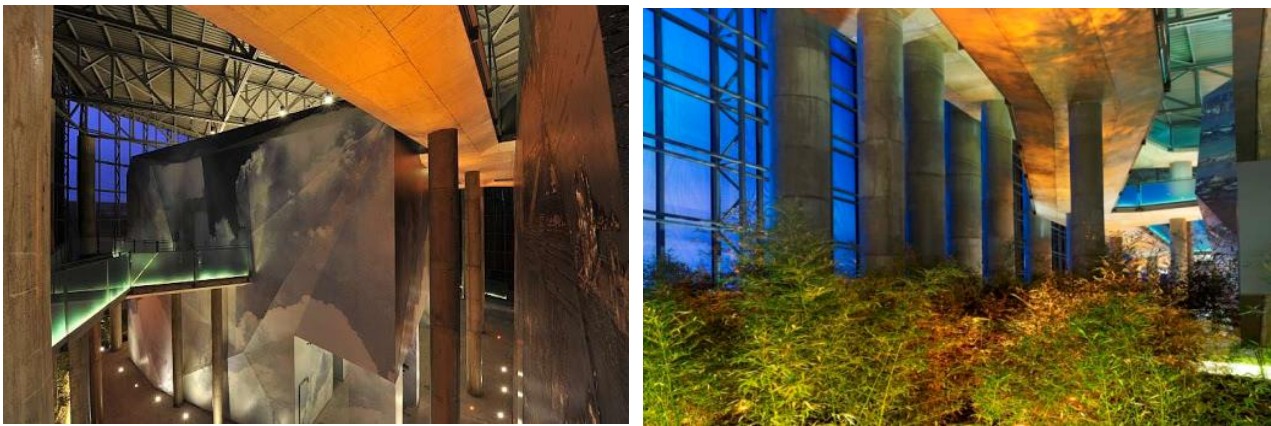

**Figure 5.** Raif Dinckok Cultural Centre, Yalova, Turkey. Object completed in 2011, architects—Emre Arolat, Gonca Pasolar and Rafat Yalmaz [8]. Photos by courtesy of emre arolat architects, Creative Commons Attribution-Share Alike 4.0 International.

The façades of the town museum building in Essen, Germany, which were built in 2009, and the city archives of Haus der Essener Geschichte—Stadtarchiv are completely covered with corten cladding. According to the authors' intention, the time-varying, maturing patina of corten steel sheets illustrates the passing of time, directly referring to the function of the building housing the documentation of the city's history. The brutal, tin block is also related to the landscape of the industrial complex of the Krupp steel plant—Kruppstadt Essen, for many years an essential condition for the city's existence (Figure 6) [9].

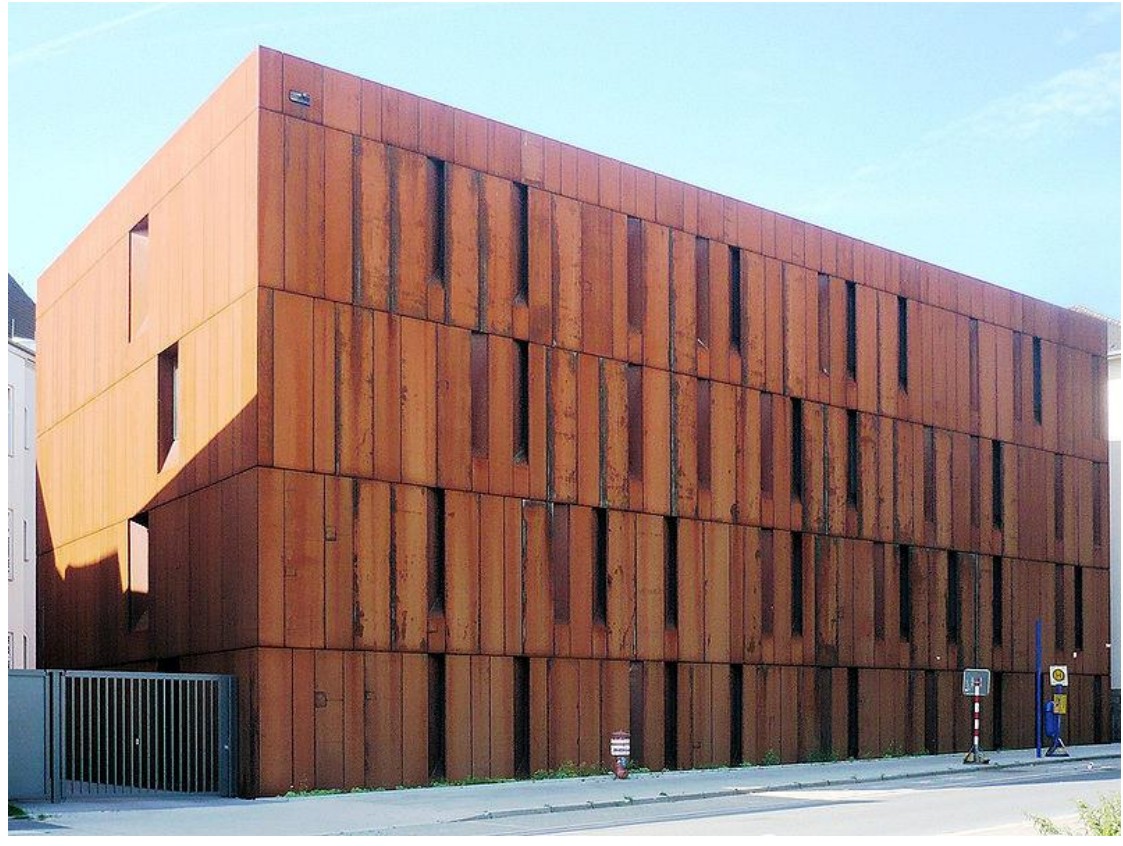

**Figure 6.** House of Essen History and City Archives, Essen, Germany. Object completed in 2009, architects—Frank Ahlbrecht, Hermann Scheidt [10] Photo—public domain.

The use of corten in architecture is good for building a symbolic aura. Architects fascinated by this material willingly use it to manifest programmatic references. Despite the aesthetic affinities with the above-mentioned projects, the building of the ECS is clearly specific. In this case, the designer took the risk of mass use of corten cladding as a finishing material in interiors with qualified acoustics. On the other hand, the unconventional use of large steel surfaces raised concerns that the interiors would show the acoustic properties of an industrial hall, and thus lose the acoustic comfort desired in public utility buildings [11].

## 3. Corten Plates as a Finishing Material

Corten steel (original name COR-TEN) is an alloy with a deliberately created corrosion layer replacing the need for a protective paint coating. It is obtained by the use of alloy additives that give the steel the desired metallurgical properties and form the corrosion layer (P, Cu, Cr, Ni, Si, Mo, Ti, V). Work on the production of corten steel began in 1933, with practical applications in 1959 [12].

In its basic applications, corten steel is a construction material. In the form of metal sheets, it is used relatively rarely, mainly due to its specific decorative value. In architecture, corten plates are mainly used as a façade covering for office and commercial buildings, cultural facilities, etc., as well as for artistic projects such as monuments, sculptures or other open-air installations. A special feature of the material is a long period of natural stabilization of the physico-chemical properties of the corrosion coating, lasting many years [13]. This is related to its change of color, as well as dusting and settling of the corrosion deposits on the surface of the ground [14]. For this reason, corten steel is mainly used in the open air. Due to the possibility of dusting, the use of corten sheet in rooms requires stabilization of the corrosion coating, e.g., by spraying with thin layer of clear varnish.

The article describes the use of corten steel cladding in the largest ECS rooms, i.e., in the winter garden and in a multi-purpose hall, along with the impact of this material on their acoustics. Corten plate is a finishing material that also dominates other ECS rooms (entrance hall, foyer, corridors). The rationale is a reference to the material uniformity of the façade and interiors, characteristic of large industrial facilities.

## 4. Use of Corten Plates in the ECS Interiors

A cladding made of 1 mm corten plate on plywood 15 mm thick was applied in all interiors of the ECS. Its sound absorption coefficient is comparable to that of plaster ($\alpha_w \approx 0.05$) [15,16]. The use of such material in the form of large, flat sheets causes the sound to be reflected in a mirror-like manner without loss of energy, which leads to a long reverberation time [17]. According to the designer's intentions, this is not considered an acoustic defect, but rather as an aid to the visual means shaping the mood of the major industrial interior.

These effects are mitigated to some extent by the use of a drawn mesh made of corten plate and perforated corten plates, moved away from the concrete wall, and the filling of the voids with mineral wool. When applied to the upper part of the wall, it preserves the homogeneity of the interior material and minimizes excessive reverberation, directing the auditory sensations of the users from the expected clear industrial monumentality to the acoustic neutrality typical of public spaces.

### 4.1. Winter Garden

The winter garden brings together the most important ECS functions, which are permanent and temporary exhibitions, reading rooms with libraries, offices, etc. They are accessible directly or through wide corridors, mezzanines with escalators and internal glass facades and window openings. As a result, the interior of the winter garden is largely fragmented, which in combination with large clusters of vegetation has a positive effect on its acoustics (Figure 7).

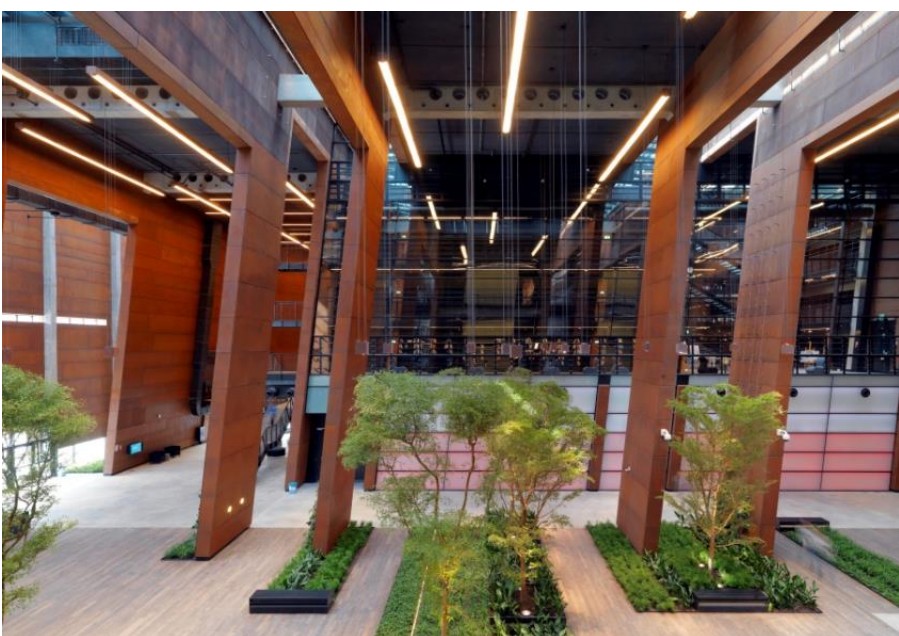

**Figure 7.** The ECS, winter garden (photo by Wojciech Kryński).

As it was said before, the interior of the winter garden is made of large, flat elements covered with metal sheets, reflecting the sound mainly in a mirror form, i.e., without dispersion and with negligible absorption. Occurring in sufficiently large groups, the vegetation has a positive effect on the acoustics of the room formed in this way by absorbing and dispersing sound [18].

There are two types of vegetation in the Winter Garden—trees growing in single pots and a green wall formed by plants climbing a frame next to construction poles. The sound absorption coefficient of detached trees is $\alpha = Gf^{1/2}$, where f is the frequency and G is a constant with values from 0.001 to 0.002 [19]. With the density of vegetation available in a room of the considered size (see Figure 4), a sound absorption coefficient $\alpha$ in the range from 0.06 to 0.18 for the frequency range f from 1 to 8 kHz can be expected. Similar values of $\alpha$ are reported in the literature for vegetation suitable for green wall formation ($\alpha$ from 0 to 0.2 for the frequency range from 200 to 1600 Hz) [20]. These are small values, but after reaching the target size by the mass of vegetation and combined with the effect of sound absorption by the air, also increasing with frequency [21], they will allow for a noticeable suppression of high sounds as intended by the designer.

Consciously used aesthetic effect should also be raised, expressed in exposing the rawness of corten combined with the softness of greenery, as well as improving air quality and bringing other benefits to various forms of activity conducted there [18]. At present, the vegetation in the winter garden is in the initial stage of growth, eventually becoming an element competing with the severity of corten. This applies at least to the lower part of the room, the total height of which is approx. 15 m.

The sound scattering and absorption effects produced in this way significantly reduce the feeling of reverberation that one might expect when entering such a large cubature. This refers to both the leading function of the winter garden, which includes meetings, exhibitions and occasional events, as well as special occurrences, such as the stage presentation of an opera (see Section 5, "Synergistic influence . . . ").

The acoustic measurements of the winter garden were made without the presence of the public (Figure 8). This is a common practice to avoid audience noises that affect the measurement result. The influence of the audience on the acoustics of a room is usually taken into account by calculation. An estimate of the audience's influence on the reverberation time using a statistical method (the so-called Sabine method) is shown in Appendix A.

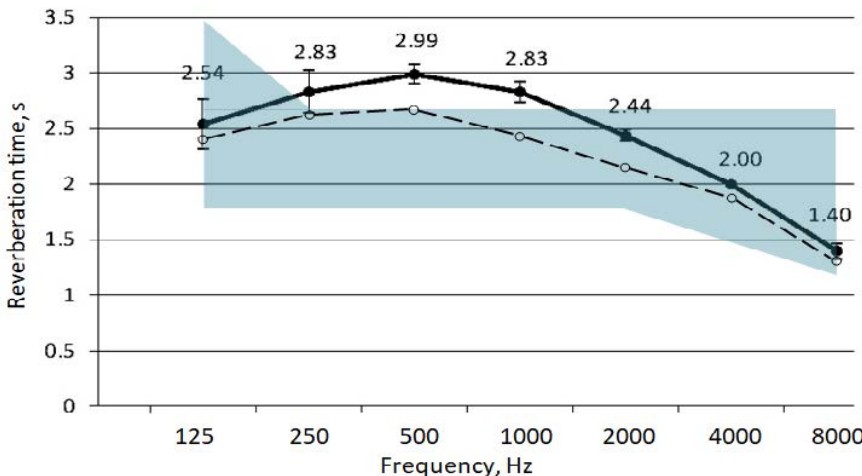

**Figure 8.** The reverberation time of the winter garden. Continuous line and numerical values: acoustical measurements (the average of 8 observation points, room without public). Measurements were made in accordance with the ISO 3382-2 standard [22], sound source—pistol shot. Dashed line—reverberation time taking into account the presence of the audience, choir and orchestra, see Appendix A. The range of recommended values of reverberation time for the occupied room is marked in blue [17].

Attention is drawn to the frequency characteristics of the reverberation time of the winter garden with the damping of the lower and upper parts of the band, due to the arrangement of the structural elements in the ceiling of the interior and the presence of large bodies of greenery, intended for further growth. In view of the varied nature of sound production, in line with the program-based multi-purpose character of the winter garden, this creates favorable conditions for the functioning of the sound system.

Dimensions of the winter garden—length × width × height: approx. 30 × 30 × 15 m, floor area—885 m², cubature, including acoustically coupled spaces—19,400 m³, surface of the corten plates—approx. 2200 m², reverberation time at a frequency of 1000 Hz—2.8 s (Figure 8).

*4.2. Multi-Purpose Hall*

The multi-purpose hall is a room with a simpler form. As with the entire building, the main tool for creating the interior architecture of the hall is the corten plates, which completely cover the parallel aligned side walls. In accordance with the accepted formula of an industrial hall, the side walls are formed from large single-piece planes. This promotes the fluttering effect of the sound as it decays. In general, this phenomenon is due to the multiple mirror reflection of sound between spaced apart parallel surfaces, e.g., between walls or ceiling and floor, and is acoustically disadvantageous. Flutter echo can also arise between specifically shaped curved surfaces with insufficient absorption [17]. This phenomenon is local in nature, i.e., it is observed only in the area including the paths of reflected waves, at a specific location of the source generating sound pulses, e.g., percussion instruments.

The estimation of flutter echo formation is based on a geometric model of sound propagation. In this model the assumption is that the obstacle that reflects the sound, in this case the side wall, is large compared to the wavelength. This can be written as $l \geq K \lambda$, where $\lambda$ is the wavelength, $l$ is the dimension of the obstacle and K is a factor dependent on the ratio between mirrored and diffused energy. K can be taken as a measure of the audibility of a flutter echo. The greater the value of K, i.e., the larger the reflecting surfaces are, the more clearly the flutter echo phenomenon is audible. At low K values, the flutter echo becomes blurry because some of the reflected sound energy takes the form of the scattered sound. With the decrease of K and the appropriate arrangement of the fragmented elements, flutter echo turns into a smooth decay of the sound, beneficial for the acoustics of the room.

In the literature, the choice of the K factor is not free of some arbitrariness. Generally, the K factor is greater than one [23,24], though some authors accept smaller values (e.g., K = 1 [25], K = 1/2 [26] or even K = 1/3 [27]). This means that even a strongly blurred flutter effect, created with a share of diffuse reflections, is treated by these authors as an acoustic defect of the room. This can be the basis for taking remedial steps, such as dividing large areas forming a room into smaller ones, corresponding to small K values. An acoustically qualified division of large areas into smaller ones leads to the creation of acoustic diffusers.

In the multi-purpose hall, the area susceptible to flutter echo formation is the stage. In accordance with the architectural concepts, the walls of the building have been tilted. In this case, the strict formal guidelines had a positive effect on the room. The walls deviate so significantly from the vertical plane such that the flutter echo formation zone is shifted above the stage, severely reducing the degree to which it is perceived (Figure 9). As mentioned above, flutter echoes are usually eliminated by positioning the walls in a suitable plane or using sound-diffusing structures [17]. In this case, in order to emphasize the form of the industrial hall and to be consistent with the formula for the whole building, the parallel position of the walls and their flat form have been preserved, at the deliberate expense of a slight inconvenience in the sphere of utility.

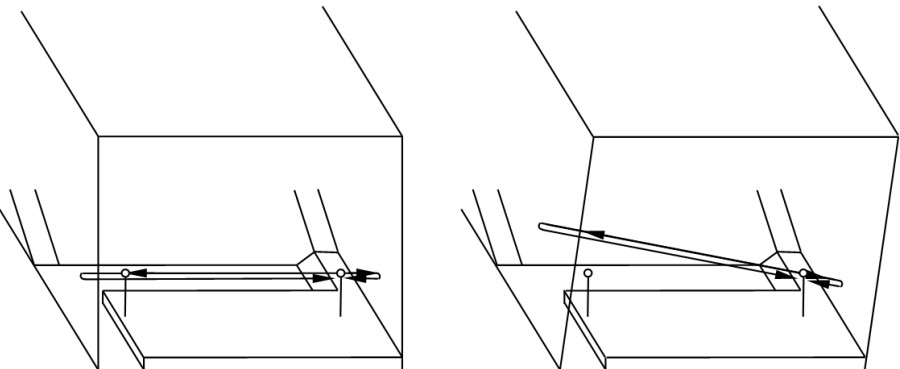

**Figure 9.** Shifting the zone of flutter echo formation (**left**) above the level of the performers' heads by deviating the side walls from the vertical (**right**).

*4.3. Acoustic Tests of the Multi-Purpose Hall*

Since the flutter echo consists of periodically repeated pulses, signal processing procedures can be used to detect their presence in the sound decay curve. Two methods are most commonly used, i.e., testing on the autocorrelation function and cepstral analysis. The autocorrelation function is a measure for the inner statistical relations of a signal and can reveal its stochastic or periodic components. In order to study the flutter echo, a property of the autocorrelation function that is called temporal diffusion was extracted [28]. Cepstral analysis is used when a highly condensed flutter echo is perceived as a coloration of sound. This method allows us to detect periodicity occurring in the sound spectrum, which is a symptom of its coloring [29].

The above-mentioned methods are an objective confirmation of subjectively unpleasant character of the reverberation, which, apart from fluttering echoes, consists of other irregularities in the sound decay curve. In many cases, a sufficient evaluation of the quality of the sound decay curve can be made using the auditory judgment. This applies especially to the audibility of the flutter echo in various types of sound production, i.e., speech, singing, musical sounds, etc., taking into account performance factors such as tempo, expression, the occurrence of pauses, type of instrumentation and others. The mentioned types of stimulation are different from the standard measurement signal. Therefore, it is helpful to include a competent judge, i.e., appropriately trained listener, in the auditory evaluation of these effects in the sound decay phenomenon.

In order to test the susceptibility of the multipurpose hall to the formation of flutter echoes, acoustic tests were carried out during the finishing works. They comprised an auditory evaluation of the degree of audibility of the flutter echo at various configurations of the absorbing material applied to the side walls (Figure 10). During the test, a comparison between the audibility of the echo to a listener positioned in the central part and at the edge of the stage was carried out, with and without sound-absorbing material. The source of the sound was a pistol shot, speech and selected musical instruments.

It was found that the inclination of the walls was a sufficient remedial step, shifting the area of the flutter echo formation above the level of the performers' heads. The level of echo was reduced to such an extent that it became unnoticeable. The echo remains slightly noticeable only at the edge of the stage, by the left side wall, which does not affect the convenience of using the room. Thanks to this, the covering of the lower parts of the walls with sound-absorbing material was avoided, which would have been clearly visible from the audience. Sound-absorbing material is applied only to the upper parts of the side walls as a means of reducing reverberation, without affecting the visual reference to the industrial character of the interior (Figure 11).

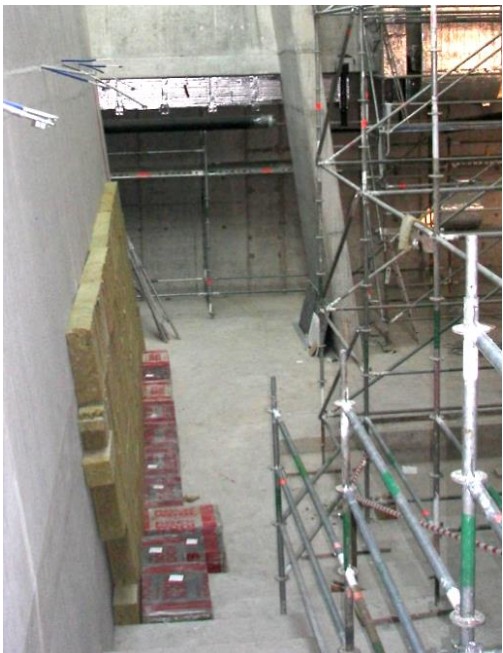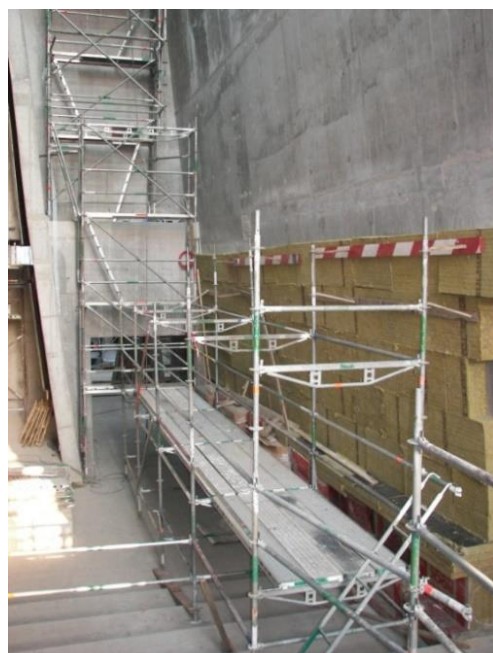

**Figure 10.** The multi-purpose hall during the acoustic experiment regarding the susceptibility of the room to the formation of the flutter echo. View from the audience on the stage, the left- and right-side walls are shown (photo by A. Kulowski).

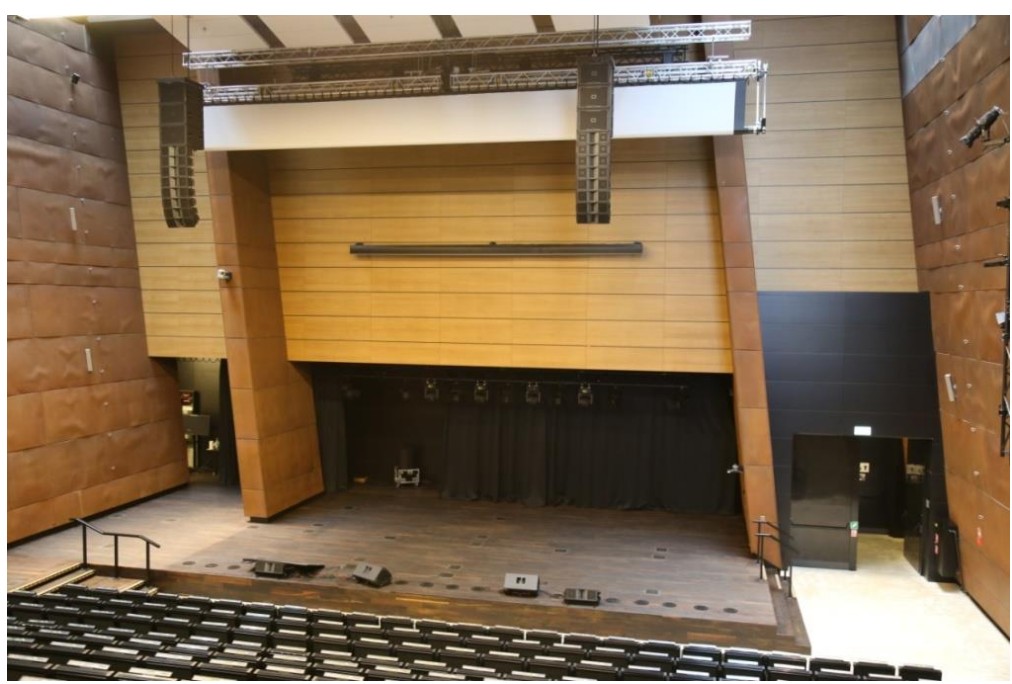

**Figure 11.** The multi-purpose hall, view from the audience on the stage (photo by Wojciech Targowski).

Dimensions of the multi-purpose hall—length × width × height—30.5 × 20.7 × 17.6 m; stage—length × width—15.3 × 9.1 m; floor area—483 m$^2$; cubature—5700 m$^3$; number of seats—433, surface of the corten plates—approx. 940 m$^2$; reverberation time at the frequency 1000 Hz—1.3 s (Figure 12).

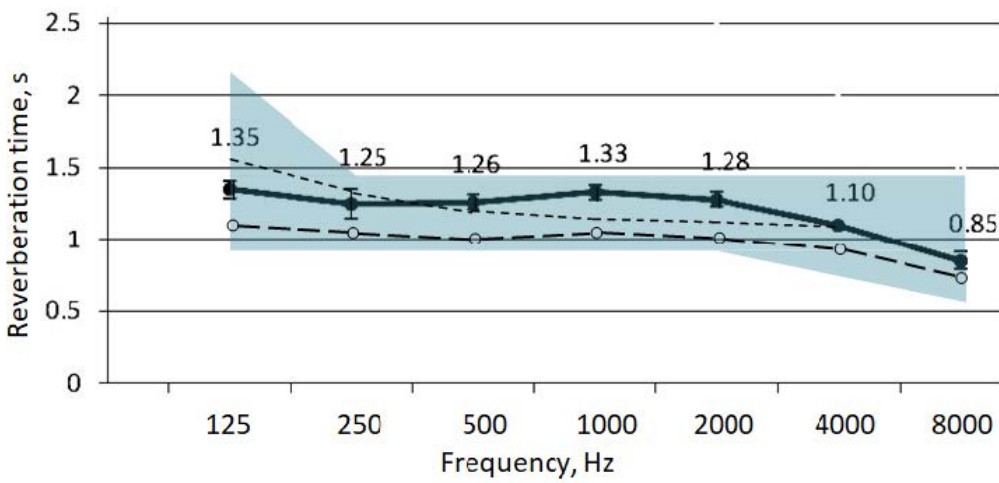

**Figure 12.** Continuous line and numerical values—measured reverberation time of the multi-purpose hall (the average of 6 observation points, room without public). The reverberation time measurements were made in accordance with the ISO 3382-2 standard [22], sound source—pistol shot. Thin dashed line—reverberation time according to the acoustic design. Thick dashed line—reverberation time in a occupied room, see Appendix A. The range of recommended values of reverberation time for occupied room is marked in blue [17].

## 5. Synergistic Influence of the Place of Presenting Art on Its Perception

The general aura of the surroundings—the spirit of the place—is important in terms of the perception of artistic events. We succumb to its charm when the architecture of the building crosses the barriers of the literary message. The emotional experience of the

space can then be transferred to the assessment and impact of the events taking place within it [30].

The industrial setting and the resulting interior acoustics take on special meaning in the context of the artistic events presented at the ECS. It acts as a confirmation of the significant influence of the place of presenting art on its perception. One example of such synergy, noticeable in almost every meeting or artistic event taking place at the ECS, is the presentation of Ludwig van Beethoven's opera "Fidelio" in the winter garden on 30 August 2017 (Figure 13). The concert was performed by native musicians along with professionally educated artists who were immigrants and refugees from countries in conflict [31]. In this context, the ECS as a place for presenting art takes on special significance. Along with the ideological message of this operatic work, it refers to still-remembered or ongoing events with significant social consequences, including the mass protests that took place here in 1981, the changes in Poland and in Europe in 1989 and the current influx of immigrants and refugees into Europe. Art becomes then a clear carrier of the idea of solidarity, connecting these events.

Unrelated to the ECS, an eloquent example of a synergistic connection between the place of traumatic events and the content of the presented piece of art is the historical performance of Wolfgang A. Mozart's "Requiem" in Sarajevo in 1994 by the orchestra and chorus of the city of Sarajevo with world-class conductor and soloists (Zubin Mehta, José Carreras) [32,33]. The concert took place in the capital of the 1984 Winter Olympics, destroyed 10 years later during the civil war in Bosnia, among the rubble of the ruined National Library.

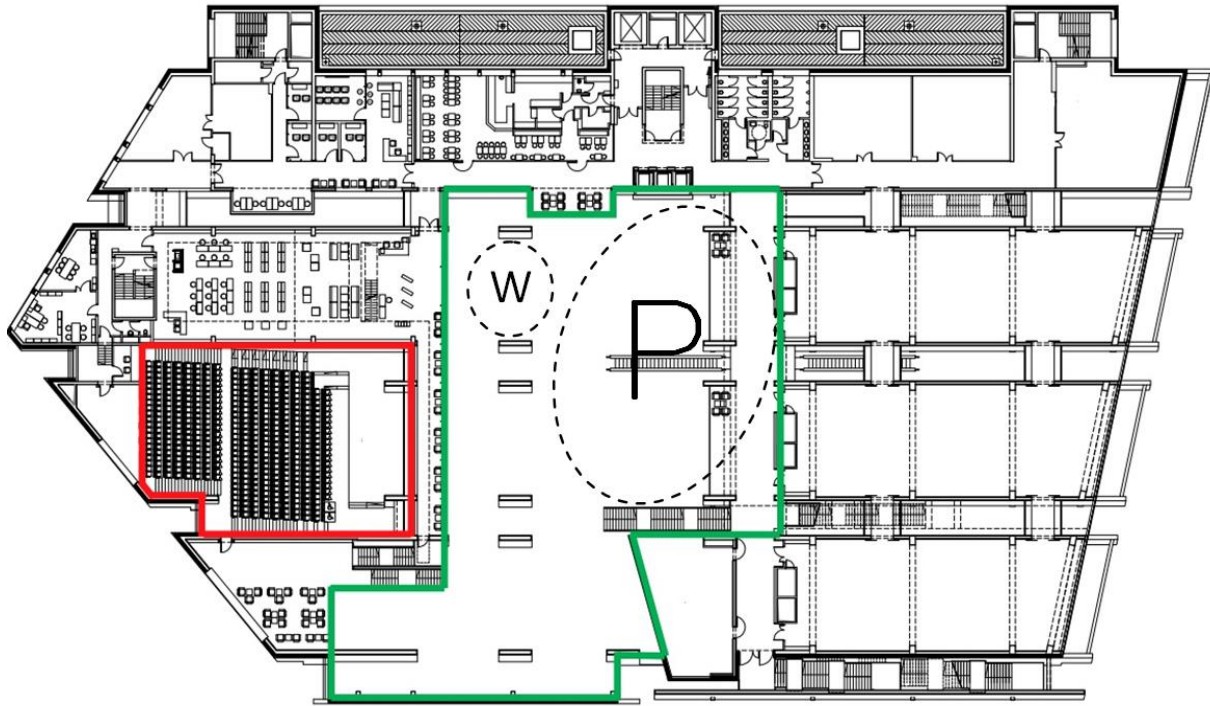

**Figure 13.** Location of the multifunctional hall and the winter garden on the ECS plan; the positioning of the performers (W) and the audience (P) during the stage performance of Ludwig van Beethoven's opera "Fidelio" in the winter garden is shown (own study).

## 6. Conclusions

The European Solidarity Centre is a public building, one in which the guiding idea of architecture plays a special role. Apart from formal procedures and numerous symbolic themes, the clear reference to industrial architecture serves to create the expected aura.

The primary means of expression are the shape of the whole, the interior layout and the uniformity of the finishing material, dominating on the façades and interiors. The building refers to the form of a large industrial factory building, with its high walls and vast, single-space assembly halls. The whole is finished with seemingly unattractive, raw in appearance, corten plates.

The main room of the ECS is a winter garden, which, together with the accompanying rooms, occupies most of the volume of the building. It was intended as a multi-purpose venue for various events involving large groups of participants. The consistent use of sheet metal as a finishing material in such a high and vast interior is a major inconvenience, leading to high reverberation, typical of industrial spaces. However, it was shown that architectural treatments can partially compensate for the demonstrable flaws. Particularly beneficial is the horizontal break in the interior, obtained due to mezzanines, the opening of the layout with wide and several meters high corridors, penetrating the whole body of the building, and the exceptionally richly designed clusters of greenery. The complementing of the whole with sound-absorbing fittings located in the upper parts of the room made it possible to achieve favorable acoustic conditions. In their current state, they allow for a variety of activities, from exhibitions, through various types of meetings, to qualified concert activities.

The multi-purpose hall, which is an important part of what the ECS program offers, was designed as a rectangular, hall-like interior, which is not conducive to good acoustics. In addition, it is completely encased in corten plate cladding. On the other hand, the tilted walls, as provided for in the project, effectively reduce the unfavorable effects and constitute the result of a desire to give the building a dynamic character, important with respect to illustrating the momentum and expansiveness of the political changes of the Solidarity era. Appropriate acoustic parameters were obtained in combination with the use of sound-absorbing materials.

The article shows that corten steel, a seemingly visually unimpressive and acoustically unsuitable finishing material, gains a new, attractive dimension through close cooperation between an architect and an acoustic specialist. The appropriate design decisions made it possible to reconcile the acoustic requirements with maintaining the clear ideological layer of the building. Furthermore, the interaction of the acoustic climate with the cultural climate of the place enhances the reception of both the values of the built space and artistic events organized in the ECS building.

**Author Contributions:** Conceptualization, W.T. and A.K.; resources, W.T.; writing—original draft preparation, A.K.; writing—review and editing, W.T. All authors have read and agreed to the published version of the manuscript.

**Funding:** This research received no external funding.

**Institutional Review Board Statement:** Not applicable.

**Informed Consent Statement:** Not applicable.

**Data Availability Statement:** Not applicable.

**Conflicts of Interest:** The authors declare no conflict of interest.

## Appendix A

**Table A1.** Calculations taking into account the influence of the audience, choir and orchestra on the reverberation time during the stage performance of Ludvig van Beethoven's opera Fidelio in the winter garden.

| No. | Acoustic Data | Octave Band, f, Hz | | | | | | |
|-----|---------------|------|------|------|------|------|------|------|
| | | 125 | 250 | 500 | 1k | 2k | 4k | 8k |
| 1 | Measured reverberation time of the room without public, T, s | 2.54 | 2.83 | 2.99 | 2.83 | 2.44 | 2.00 | 1.40 |
| 2 | Sound absorption or the room without public, $A = 0.161 \times 19{,}400/T$, m$^2$ | 1230 | 1104 | 1045 | 1104 | 1280 | 1562 | 2231 |
| 3 | Sound abs. coefficient, person sitting in an upholstered chair, $\alpha_1$ [17] | 0.16 | 0.35 | 0.42 | 0.47 | 0.52 | 0.53 | 0.54 |
| 4 | Sound absorption of 400-seat audience as in No. 3, 2 persons per 1 m$^2$, $A_1 = 200 \times \alpha_1$, m$^2$ | 32 | 70 | 84 | 94 | 104 | 106 | 108 |
| 5 | Sound absorption of an adult in light clothing, $A_{20}$, m$^2$ [17] | 0.60 | 0.95 | 1.06 | 1.08 | 1.08 | 1.08 | 1.08 |
| 6 | Sound absorption of a 30-person choir as in No. 5, $A_2 = 30 \times A_{20}$, m$^2$ | 18.0 | 28.5 | 31.8 | 32.4 | 32.4 | 32.4 | 32.4 |
| 7 | Sound absorption of musician with instrument, $A_{30}$, m$^2$ [17] | 0.12 | 0.24 | 0.59 | 0.98 | 1.12 | 1.12 | 1.12 |
| 8 | Sound abs. of a 40-person orchestra as in No. 7, $A = 40 \times A_{30}$, m$^2$ | 4.8 | 9.6 | 23.6 | 39.2 | 44.8 | 44.8 | 44.8 |
| 9 | Sound absorption of the occupied room, $A' = A + A_1 + A_2 + A_3$, m$^2$ | 1285 | 1212 | 1184 | 1270 | 1461 | 1745 | 2416 |
| 10 | Reverberation time of the occupied room, $T' = 0.161 \times 19{,}400/A'$, s | 2.43 | 2.57 | 2.64 | 2.46 | 2.14 | 1.80 | 1.30 |

**Table A2.** Calculations taking into account the influence of the audience on the reverberation time in the multi-purpose hall.

| No. | Acoustic Data | Octave Band, f, Hz | | | | | | |
|-----|---------------|------|------|------|------|------|------|------|
| | | 125 | 250 | 500 | 1k | 2k | 4k | 8k |
| 1 | Measured reverberation time of the room without public, T, s | 1.35 | 1.25 | 1.26 | 1.33 | 1.28 | 1.10 | 0.85 |
| 2 | Sound absorption or the room without public, $A = 0.161 \times 5700/T$, m$^2$ | 680 | 734 | 738 | 690 | 717 | 834 | 1080 |
| 3 | Sound absorption coefficient of a person sitting in lightly upholstered theater chairs, $\alpha_4$ [17] | 0.56 | 0.68 | 0.79 | 0.83 | 0.86 | 0.86 | 0.86 |
| 4 | Sound absorption of 433-seat audience as in No. 3, 2 persons per 1 m$^2$, $A_4 = 216.5 \times \alpha_4$, m$^2$ | 121 | 147 | 171 | 179 | 186 | 186 | 186 |
| 5 | Sound absorption of the occupied room, $A' = A + A_4$, m$^2$ | 801 | 881 | 909 | 869 | 903 | 1020 | 1266 |
| 6 | Reverberation time of the occupied room, $T' = 0.161 \times 5700/A'$, s | 1.14 | 1.04 | 1.00 | 1.06 | 1.02 | 0.90 | 0.72 |

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
