# Peer review of "Influence of the Widespread Use of Corten Plate on the Acoustics of the European Solidarity Centre Building in Gdańsk"

_buildings, doi:10.3390/buildings11030133_

Round 1
Reviewer 1 Report
In abstract.
This paper describes two rooms Winter garden and Multi-purpose hall, but if you read the abstract just refers one room.
the use of the abbreviation ECS , which must be defined.
In introducing, it would be interesting to introduce some similar examples of buildings for the same purpose.
In line 110-111 It is necessary to introduce a reference “3-year period of natural stabilization of the physical and chemical properties of the corrosion coating”.
In line 126. Introduce a reference about “ This lead to significant … dispersion”.
Line 140, introduce characteristics or reasons why vegetation has a positive effect and reference.
In winter garden, the study of reverberation in low frequency 250Hz, 500 Hz and 1000Hz exceeds the recommended vale , Any recommendation to reduce this time? any action to reduce it
In figure 5, empty room?? What does it mean?? Like figure 4 or maybe the study was carried out during construction
In figure 9 same question, empty room like figure 7 or figure 8.
Line 183-194, address the estimation of flutter echo but which value of K has been used or is considered the most appropriate
Title of 3.3 must be changed , por example Acoustic test in Multi-purpose hall.
Line 272 , personal feelings should be avoided
Author Response
REVIEWER 1
|
Comments of the reviewer |
Author's response (see corrections marked in red in the manuscript) |
|
In abstract, this paper describes two rooms Winter garden and Multi-purpose hall, but if you read the abstract just refers one room. |
The correction is made in the abstract. |
|
the use of the abbreviation ECS , which must be defined. |
The correction is made in the abstract. |
|
In introducing, it would be interesting to introduce some similar examples of buildings for the same purpose. |
The examples of buildings are described in the new point "2. Examples of widespread use of corten cladding in public utility buildings" . |
|
In line 110-111,i is necessary to introduce a reference “3-year period of natural stabilization of the physical and chemical properties of the corrosion coating”. |
The reference on the stabilization of the corten properties is added. |
|
In line 126. Introduce a reference about “ This lead to significant … dispersion”. |
The comment is added. |
|
Line 140, introduce characteristics or reasons why vegetation has a positive effect and reference. |
The comment and the reference on the influence of vegetation is added. |
|
In winter garden, the study of reverberation in low frequency 250Hz, 500 Hz and 1000Hz exceeds the recommended value , Any recommendation to reduce this time? any action to reduce it |
The comment on the hall's acoustics without public is added. The audience's influence on the reverberation time is calculated, the results are marked in Fig. 8 and 12. The details of calculations are shown in the Appendix. |
|
In figure 5, empty room?? What does it mean?? Like figure 4 or maybe the study was carried out during construction |
|
|
In figure 9 same question, empty room like figure 7 or figure 8. |
|
|
Line 183-194, address the estimation of flutter echo but which value of K has been used or is considered the most appropriate |
Notes on the interpretation of K are added. |
|
Title of 3.3 must be changed , por example Acoustic test in Multi-purpose hall. |
The title is corrected. |
|
Line 272 , personal feelings should be avoided |
The author, and at the same time the main designer of the ECS, allows himself to retain the right to express personal feelings about this place, as they became the inspiration for design decisions regarding the ECS. |

Reviewer 2 Report
The paper shows an interesting research topic about the use of corten steel as finishing material on façades and interiors.
However some major revisions must be performed on the paper before the publication:
1-a more accurate description of the state of the art with more references is necessary;
2- a more accurate description of the measurement set-up must be inserted;
3- a comparison between the corten steel and other similar material is necessary also taking into account comparable architectural examples.
Author Response
REVIEWER 2
|
Comments of the reviewer |
Author's response (see corrections marked in red in the manuscript) |
|
1. A more accurate description of the state of the art with more references is necessary; |
New point "2. Examples of widespread use of corten cladding in public utility buildings" is added. |
|
2. A more accurate description of the measurement set up must be inserted; |
The comment and reference to the EN ISO 3382-2 standard is added. |
|
3. A comparison between the corten steel and other similar material is necessary also taking into account comparable architectural examples. |
A comparison between the acoustical properties of corten steel and plaster is given in Point 4. Comparable architectural examples are discussed in Point 2. |
Reviewer 3 Report
The publication describes in a way the interplay between acoustics and architecture. This is valuable research subject! However, the issue is not approached scientifically and systematically here. Rather, a single example is treated without a new scientific approach being apparent. This may be very interesting for an architectural journal or a technical journal. However, the requirements of a scientific journal are not met here.
Concerning Acoustics: Overall, it is rather the application of known procedures and rules of acoustics. There is nothing wrong with this, and it is actually exemplary for acoustical consulting. But it is not appropriate for a scientific publication.
Incidentally, one clearly misses the scientific-technical depth in the acoustic parts of the publication. Also for an engineer's report, the usual essential "details" are missing.
A few examples will illustrate the lack of depth in the following. However, remedying the deficiencies would unfortunately not significantly improve the scientific content of the publication.
- The references to non-English literature clearly complicate the understanding. Here, a short summary of these contents and further clarifying literature references would be absolutely necessary.
- There is no information whether the reverberation times were measured according to the usual standard.
- 3.3 Acoustic tests: Only the results are presented here. The methodology of the tests, the procedures for statistical evaluation, etc. are missing. It can be assumed that these are only simple listening tests. This would be o.k. in itself, but even then a better description of the challenges would be desirable - preferably also connected with acoustic measurements.
- Figure 9. Apart from the hint whether it was measured according to the standard, the calculation for the state occupied by the audience is missing here.
Of course, the scientific criticism of the paper does not mean a lack of respect for the architectural project and the good cooperation between architecture and acoustics.
Author Response
REVIEWER 3
|
Comments of the reviewer |
Author's response (see corrections marked in red in the manuscript) |
|
The references to non-English literature clearly complicate the understanding. Here, a short summary of these contents and further clarifying literature references would be absolutely necessary. |
The questioned references are removed. |
|
There is no information whether the reverberation times were measured according to the usual standard. |
The comment and reference to the EN ISO 3382-2 standard is added. |
|
3.3 Acoustic tests: Only the results are presented here. The methodology of the tests, the procedures for statistical evaluation, etc. are missing. It can be assumed that these are only simple listening tests. This would be o.k. in itself, but even then a better description of the challenges would be desirable - preferably also connected with acoustic measurements. |
The methodology of measuring the presence of echo flutter echo in the sound decay phenomenon and arguments for performing listening tests are described. |
|
Figure 9. Apart from the hint whether it was measured according to the standard, the calculation for the state occupied by the audience is missing here. |
The reasons for measuring the reverberation time in a hall without an audience are given. The audience's influence on the reverberation time is calculated, the results are marked in Fig. 8 and 12. The details of calculations are shown in the Appendix. |
Round 2
Reviewer 1 Report
The authors have done a good job addressing all the comments and suggestions provided in the first round of revision.
However, some minor issues should be also considered before the paper deserves publication.
1- Some mistakes with garden in line 10 and line 161. WARDEN--> GARDEN.
2 - In figure 3, is entirely covered with--> it is possible that you are referring to both facades. If I am right, they should be changed. "They are entirely covered...
3- In line 142, acoustic neutrality what does it mean??
4- Line 168 (αw≈0.05). Please give a reference.
5 – this leads to small dispersion and a long reverberation time. Please give any reference
6- Lines 188-190 . “Occurring in sufficiently large groups, the vegetation has a positive effect on the acoustics of the room formed in this way by absorbing and dispersing sound. “
Please give any reference where this shows. Or the reference is [14].
7- Appendix. Please use the same digit number after the decimal point. There are some number with 1 , other with 2, other with no decimal.
Author Response
REVIEWER 1
|
Comments of the reviewer |
Author's response (after changing the line numbering, the places for corrections are also shown in the "Track changes" function in Microsoft Word) |
|
1- Some mistakes with garden in line 10 and line 161. WARDEN--> GARDEN. |
Typing errors have been corrected. |
|
2 - In figure 3, is entirely covered with--> it is possible that you are referring to both facades. If I am right, they should be changed. "They are entirely covered... |
The caption for Figure 3 has been corrected, the term "completely covered" has been removed. |
|
3- In line 142, acoustic neutrality what does it mean?? |
The last sentence of point 1: The phrase "acoustic neutrality" has been replaced by "acoustic comfort" with commentary and reference to the literature. |
|
4- Line 168 (αw≈0,05). Please give a reference. |
Second sentence of point 4: References are given to the symbol αw and the value 0,05. |
|
5 – this leads to small dispersion and a long reverberation time. Please give any reference |
Third sentence of point 4: The phrase "small dispersion" has been removed, reference is made to the long reverberation time. |
|
6- Lines 188-190 . “Occurring in sufficiently large groups, the vegetation has a positive effect on the acoustics of the room formed in this way by absorbing and dispersing sound. “ Please give any reference where this shows. Or the reference is [14] |
In point 4.1, a new paragraph has been added regarding the sound absorption coefficient of vegetation and the expected impact of greenery on the acoustics of the winter garden, with references to the literature. |
|
7- Appendix Please use the same digit number after the decimal point. There are some number with 1, other with 2, other with no decimal.
|
Appendix Individual values are given with a different number of decimal places, as they relate to different parameters. The sound absorption coefficient α is given with 2 decimal points (the accuracy 1%), the increment of the equivalent absorption area is given with 1 decimal point, the absorption area of the whole room has been rounded to integer values. |

Reviewer 2 Report
Dear Authors,
now the article is well written.
Any changes are necessary.
Author Response
REVIEWER 2
|
Comments of the reviewer |
Author's response |
|
Now the article is well written. Any changes are necessary. |
The authors thank the reviewer for the positive assessment of the article. |

Reviewer 3 Report
10, 12, 161
winter warden > winter garden
142
It is not clear what is meant by "acoustic neutrality" as it is not a common term in room acoustics. Either it has to be explained what is really meant, or else the term has to be replaced. It can be assumed that what is meant is: The acoustic conditions would then not meet the requirements of the use.
168
Please give a reference to αw: ISO 11654
169
small sound dispersion > almost no sound scattering
189
"vegetation has a positive effect on the acoustics of the room formed in this way by absorbing and dispersing sound."
Please provide a reference that shows how great the sound absorption coefficient of plants is. It would also be desirable to provide an estimate with a rough calculation showing how large the effect finally is. It can be estimated that the effect is rather small. It should therefore be transparently pointed out here that the effect is rather small.
Appendix
It is unreasonable to give the Equivalent Absorption Area A to one digit after the decimal point. An accuracy of 0.1 m2 is suggested, which is nonsense
Author Response
REVIEWER 3
|
Comments of the reviewer |
Author's response (after the line renumbering, the places for corrections are also shown in the "Track changes" function in Microsoft Word) |
|
10, 161 |
Typing errors have been corrected |
|
142 |
The last sentence of Point 1: The phrase "acoustic neutrality" has been replaced by "acoustic comfort" with commentary and reference to the literature. |
|
168 |
Second sentence of point 4: References are given to the symbol αw and the value 0,05. |
|
169 |
Third sentence of point 4: The phrase "small dispersion" has been removed, reference is made to the long reverberation time. |
|
189 |
In point 4.1, a new paragraph has been added regarding the sound absorption coefficient of vegetation and the expected impact of greenery on the acoustics of the winter garden, with references to the literature. |
|
Appendix It is unreasonable to give the Equivalent Absorption Area A to one digit after the decimal point. An accuracy of 0.1 m2 is suggested, which is nonsense |
Appendix Equivalent absorption area A of the room has been rounded to integer values. |
